# Instance-Dependent Label-Noise Learning under Structural Causal Models

**Yu Yao**[1]    **Tongliang Liu**[1][†]   **Mingming Gong**[2]
**Bo Han**[3]     **Gang Niu**[4]    **Kun Zhang**[5]
[1]TML Lab, University of Sydney; [2]University of Melbourne; [3]Hong Kong Baptist University;
[4]RIKEN AIP; [5]Carnegie Mellon University

## Abstract

Label noise generally degenerates the performance of deep learning algorithms because deep neural networks easily overfit label errors. Let $X$ and $Y$ denote the instance and clean label, respectively. When $Y$ is a cause of $X$, according to which many datasets have been constructed, e.g., *SVHN* and *CIFAR*, the distributions of $P(X)$ and $P(Y|X)$ are generally entangled. This means that the unsupervised instances are helpful to learn the classifier and thus reduce the side effect of label noise. However, it remains elusive on how to exploit the causal information to handle the label-noise problem. We propose to model and make use of the causal process in order to correct the label-noise effect. Empirically, the proposed method outperforms all state-of-the-art methods on both synthetic and real-world label-noise datasets.

## 1 Introduction

Learning with noisy labels can be dated back to [1] and has recently drawn a lot of attention [15, 19, 11, 10, 28]. In real life, large-scale datasets are likely to contain label noise. It is partly because that many cheap but imperfect data collection methods such as crowd-sourcing and web crawling are widely used to build large-scale datasets. Training with such data usually lead to poor generalization abilities of deep neural networks because they can memorize noisy labels [2, 33].

To improve the generalization ability of training with noisy labels, one family of existing label-noise learning methods is to model how the label noise was generated [17, 19, 27, 34, 14]. Specifically, these methods try to reveal the transition relationship from clean labels to noisy labels of instances, i.e., the distribution $P(\tilde{Y}|Y, X)$, where $\tilde{Y}$, $Y$ and $X$ are the random variables for the noisy label, latent clean label, and instance, respectively. The advantage of modelling label noise is that given only the noisy data, when the transition relationship is identifiable, classifiers can be learned to converge to the optimal ones defined by the clean data, with theoretical guarantees. However, the transition relationship is not identifiable in general. To make it identifiable, various assumptions have been made on the transition relationship. For example, Natarajan et al. [17] assume that the transition relationship is instance independent, i.e., $P(\tilde{Y}|Y, X) = P(\tilde{Y}|Y)$; Xia et al. [30] assume that the $P(\tilde{Y}|Y, X)$ is dependent on different parts of an instance. Cheng et al. [7] assume that the label noise rates are upper bounded. In practice, these assumptions may not be satisfied and are generally hard to be verified given noisy data alone.

Inspired by causal learning [20, 25, 21, 23], we provide a causal perspective of label-noise learning method named *CausalNL*. We exploit the causal information to help identifiability of the transition matrix $P(\tilde{Y}|Y, X)$ other than making assumptions directly on the transition relationship.

---

[†]Correspondence to Tongliang Liu (tongliang.liu@sydney.edu.au).

35th Conference on Neural Information Processing Systems (NeurIPS 2021).

Specifically, we assume that the data containing instance-dependent label noise is generated according to the causal graph in Fig. 1. For example, for the Street View House Numbers (SVHN) dataset [18], $X$ represents the image containing the digit; $Y$ represents the clean label of the digit shown on the plate; $Z$ represents the latent variable that captures the information affecting the generation of the images, e.g., orientation, lighting, and font style. Here $Y$ is naturally a cause of $X$ and the causal generative process can be described in the following way. First, the house plate is generated according to the street number and attached to the front door. Then, the house plate is captured by a camera (installed in a Google street view car) to form $X$, taking into account of other factors such as illumination and viewpoint. Finally, the images containing house numbers are collected and relabeled to form the dataset. Let us

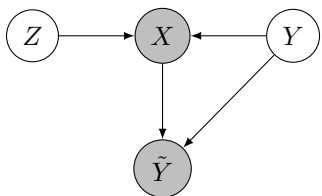

Figure 1: A graphical causal model which reveals a generative process of the data which contains instance-dependent label noise, where the shaded variables are observable and the unshaded variables are latent.

denote the annotated label by the noisy label $\tilde{Y}$ as the annotator may not be always reliable, especially when the dataset is very large but the budget is limited. During the annotation process, the noisy labels were generated according to both the images and the range of predefined digit numbers. Hence, both $X$ and $Y$ are causes of $\tilde{Y}$. Note that most existing image datasets are collected with the causal relationship that $Y$ causes $X$. For example, see the widely used *FashionMNIST* and *CIFAR*. When we synthesize instance-dependent label noise based on them, we will have the causal graph illustrated in Fig. 1. Note also that some datasets are generated with the causal relationship that $X$ causes $Y$. Other than using domain knowledge, the different causal relationships can be verified by employing causal discovery [26, 25, 21, 37].

When the latent clean label $Y$ is a cause of $X$, $P(X)$ will generally contain some information about $P(Y|X)$. This is because, under such a generative process, the distributions of $P(X)$ and $P(Y|X)$ are entangled [22, 39]. To help estimate $P(Y|X)$ with $P(X)$, we make use of the causal generative process to estimate $P(X|Y)$, which directly benefits from $P(X)$ by generative modeling. The modeling of $P(X|Y)$ in turn encourages the identifiability of the transition relationship and helps learn $P(Y|X)$. For example, in Fig. 2(a), we have added instance-dependent label-noise with a rate 45% (i.e., IDLN-45%) to the MOON dataset and employed different methods [10, 36] to solve the label-noise learning problem. As illustrated in Fig. 2(b) and Fig. 2(c), previous methods fail to infer clean labels. In contrast, by constraining the conditional distribution of the instances, i.e., restricting the data of each class to be on a manifold by setting the dimension of the latent variable $Z$ to be 1-dimensional, the label transition as well as the clean labels can be successfully recovered (by the proposed method), which is showed in Fig. 2(d). It is worth noting that the idea of finding $P(Y|X)$ by modeling $P(Y)$ and $P(X|Y)$ instead has been exploited in the context of domain adaptation; for instance, it inspires target shift, (generalized) conditional shift and other settings for domain adaptation [38, 9]

Specifically, to make use of the causal graph to contribute to the identifiability of the transition matrix, we propose a causally inspired deep generative method, which models the causal structure with all the observable and latent variables, i.e., the instance $X$, the noisy label $\tilde{Y}$, the latent feature $Z$, and the latent clean label $Y$. The proposed generative model captures the variables' relationship indicated by the causal graph. Furthermore, built on the variational autoencoder (VAE) framework [12], we build an inference network which could efficiently infer the latent variables $Z$ and $Y$ when maximising the marginal likelihood $p(X, \tilde{Y})$ on the given noisy data. In the decoder phase, the data will be reconstructed by exploiting the conditional distribution of instances $P(X|Y, Z)$ and the transition relationship $P(\tilde{Y}|Y, X)$, i.e.,

$$p_\theta(X, \tilde{Y}) = \int_{z,y} P(Z = z)P(Y = y)p_{\theta_1}(X|Y = y, Z = z)p_{\theta_2}(\tilde{Y}|Y = y, X)\mathrm{d}z\mathrm{d}y$$

will be exploited, where $\theta := (\theta_1, \theta_2)$ are the parameters of the causal generative model (more details can be found in Section 3). Ay a high level, according to the equation, given the noisy data and the distributions of $Z$ and $Y$, constraining $p_{\theta_1}(X|Y, Z)$ will also greatly reduce the uncertainty of $p_{\theta_2}(\tilde{Y}|Y, X)$ and thus contribute to the identifiability of the transition matrix. Note that adding a constraint on $p_{\theta_1}(X|Y, Z)$ is natural; for example, images often have a low-dimensional manifold [3]. We can restrict $P(Z)$ to fulfill the constraint on $p_{\theta_1}(X|Y, Z)$. By exploiting the causal structure and

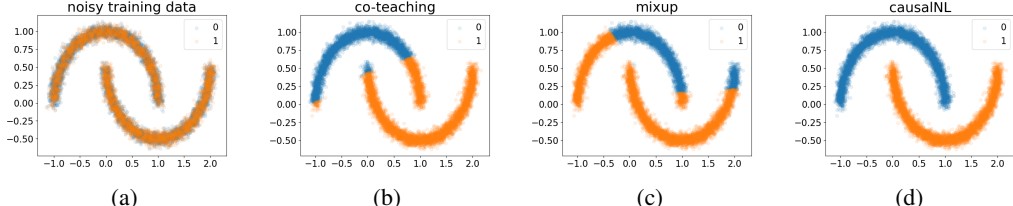

Figure 2: (a) An illustration of the MOON training dataset which contains $45\%$ of instance-dependent label noise. Different instances have different noise rates which are randomly generated according to Xia et al. [30]. (b)-(d) The illustration of the classification performance of co-teaching, mixup, and our method, respectively.

the constraint on instances to better model label noise, the proposed method significantly outperforms the baselines. When the label noise rate is large, the superiority is evidenced by a large gain in the classification performance.

The rest of the paper is organized as follows. In Section 2, we briefly review the background knowledge of label-noise learning and causality. In Section 3, we formulate our method, named *CausalNL*, and discuss how it helps to learn a clean classifier, followed by the implementation details. Experimental validations are provided in Section 4. Section 5 concludes the paper.

## 2 Noisy Labels and Causality

In this section, firstly, we introduce how to model label noise. Then, we introduce the structural causal model and discuss how to exploit the model to encourage the identifiability of the transition relationship and help learn the classifier.

**Transition Relationship**   To build a statistically consistent classifier that converges to the optimal classifier defined on clean data by only employing noisy data, the transition relationship $P(\tilde{Y}|Y, X)$ has to be identified. Given an instance, the conditional distribution can be written in an $C \times C$ matrix which is called the transition matrix [19, 29, 30], where $C$ represents the number of classes. Specifically, for each instance $x$, there is a transition matrix $T(x)$. The $ij$-th entry of the transition matrix is $T_{ij}(x) = P(\tilde{Y} = i|Y = j, X = x)$, which represents the probability that the instance $x$ with the clean label $Y = j$ will have a noisy label $\tilde{Y} = i$.

The transition matrix has been widely studied to build statistically consistent classifiers, because the clean class posterior distribution $P(\boldsymbol{Y}|x) = [P(Y = 1|X = x), \dots, P(Y = C|X = x)]^\top$ can be inferred by using the transition matrix and the noisy class posterior $P(\tilde{\boldsymbol{Y}}|x) = [P(\tilde{Y} = 1|X = x), \dots, P(\tilde{Y} = C|X = x)]^\top$, i.e., we have $P(\tilde{\boldsymbol{Y}}|x) = T(x)P(\boldsymbol{Y}|x)$. Specifically, the transition matrix has been used to modify loss functions to build risk-consistent estimators; see, e.g., [8, 19, 35, 28], and has been used to correct hypotheses to build classifier-consistent algorithms; see, e.g., [17, 24, 19]. Moreover, the state-of-the-art statically inconsistent algorithms [11, 10] also use diagonal entries of the transition matrix to help select reliable examples used for training.

However, the distribution $P(\tilde{Y}|Y, X)$ is not generally identifiable [28]. To make it identifiable, one has to resort to additional assumptions. The most widely used assumption is that given clean label $Y$, the noisy label $\tilde{Y}$ is conditionally independent of instance $X$, i.e., $P(\tilde{Y}|Y, X) = P(\tilde{Y}|Y)$. Under such an assumption, the transition relationship $P(\tilde{Y}|Y)$ can be successfully identified with the anchor point assumption [15, 34, 14]. However, in the real-world scenarios, this assumption may be hard to satisfied. Although $P(\tilde{Y}|Y)$ can be used to approximate $P(\tilde{Y}|Y, X)$, the approximation error can be large in many cases. As for the efforts to model the instance-dependent transition matrix directly, existing methods rely on rather strong assumptions, e.g., the bounded noise rate assumption [7], the part-dependent label noise assumption [30], and the requirement of additional information about the transition matrix [4]. Although the assumptions help the methods achieve superior performance empirically, they are generally difficult to verify or fulfill, limiting their applications in practice.

**Structural Causal Model**   Motivated by the limitation of the current methods, we provide a new causal perspective to learn the identifiable of instance-dependent label noise model. Here we briefly

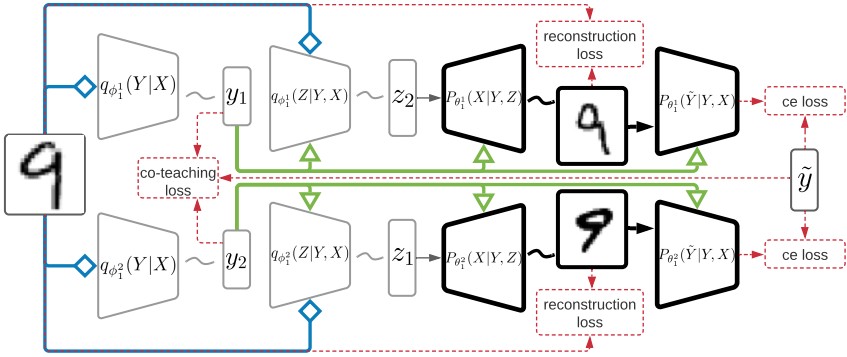

Figure 3: A working flow of our method.

introduce some background knowledge of causality [25] used in this paper. A structural causal model (SCM) consists of a set of variables connected by a set of functions. It represents a flow of information and reveals causal relationships among all the variables, providing a fine-grained description of the data generation process. The causal structure encoded by SCMs can be represented as a graphical casual model as shown in Fig. 1, where each node is a variable and each edge is a function involving noise. The SCM corresponding to the graph in Fig. 1 can be written as

$$Z = \epsilon_Z, \ Y = \epsilon_Y, \ X = f(Z, Y, \epsilon_X), \ \tilde{Y} = f(X, Y, \epsilon_{\tilde{Y}}), \tag{1}$$

where $\epsilon_Z$, $\epsilon_Y$, $\epsilon_X$ and $\epsilon_{\tilde{Y}}$ are independent exogenous variables, and they sometimes also called *error* variables. For example, $\epsilon_X$ are an error variable for $X$, which is responsible for any difference between the actual value of $X$ and the value predicted on the basis of $Z$ and $Y$ alone. Each equation specifies a distribution of a variable conditioned on its parents (which are an empty set for root cause variables in the graph).

By making use of the SCM, the benefit of the instances to learning the classifier can be clearly explained. Specifically, the instance $X$ is a function of its label $Y$ and latent feature $Z$, which means that the instance $X$ is generated from $Y$ and $Z$. Therefore $X$ must contains information about its clean label $Y$ and latent feature $Z$. That is the reason that $P(X)$ can help identify $P(Y|X)$ and also $P(Z|X)$. However, since we do not have clean labels, it is hard to fully identify $P(Y|X)$ from $P(X)$ in the unsupervised setting. For example, on the MOON dataset shown in Fig. 2 , we can possibly discover the two clusters by enforcing the manifold constraint, but it is impossible to see which class each cluster belongs to. We show in the following that we can make use of the property of $P(X|Y)$ to help model label noise, i.e., encourage the identifiability of the transition relationship, thereby learning a better classifier.

Specifically, under the Markov condition [20], which intuitively means the independence of exogenous variables, the joint distribution $P(\tilde{Y}, X, Y, Z)$ specified by the SCM can be factorized as follows.

$$P(X, \tilde{Y}, Y, Z) = P(Y)P(Z)P(X|Y, Z)P(\tilde{Y}|Y, X). \tag{2}$$

This motivates us to extend VAE [12] to perform inference in our causal model to fit the noisy data in the next section. In the decoder phase, given the noisy data and the distributions of $Z$ and $Y$, adding a constraint on $P(X|Y, Z)$ will reduce the uncertainty in the distribution $P(\tilde{Y}|Y, X)$. In other words, modeling of $P(X|Y, Z)$ will encourage the identifiability of the transition relationship and thus better model label noise. Since $P(\tilde{Y}|Y, X)$ functions as a bridge to connect the noisy labels to clean labels, we accordingly can better learn $P(Y|X)$ or the classifier by only using the noisy data.

There are normally two ways to add constraints on the instances, i.e., assuming a specific parametric generative model or introducing prior knowledge of the instances. In this paper, since we mainly study the image classification problem with noisy labels, we focus on the manifold property of images and apply the low-dimensional manifold constraint to the instances.

## 3 Causality Captured Instance-Dependent Label-Noise Learning

In this section, we propose a structural generative method which captures the causal relationship and utilizes $P(X)$ to help identify the label-noise transition matrix, and therefore, our method leads to a better classifier that assigns more accurate labels.

## 3.1 Variational Inference under the Structural Causal Model

To model the generation process of noisy data and to approximate the distribution of the noisy data, our method is designed to follow the causal factorization (see Eq. 2). Specifically, our model contains two decoder networks which jointly model a distribution $p_\theta(X, \tilde{Y}|Y, Z)$ and two encoder (inference) networks which jointly model the posterior distribution $q_\phi(Z, Y|X)$. Here we discuss each component of our model in detail.

Let the two decoder networks model the distributions $p_{\theta_1}(X|Y, Z)$ and $p_{\theta_2}(\tilde{Y}|Y, X)$, respectively. Let $\theta_1$ and $\theta_2$ be learnable parameters of the distributions. Without loss of generality, we set $p(Z)$ to a standard normal distribution and $p(Y)$ to a uniform distribution. Then, modeling the joint distribution in Eq. 2 boils down to modeling the distribution $p_\theta(X, \tilde{Y}|Y, Z)$, which is decomposed as follows:

$$p_\theta(X, \tilde{Y}|Y, Z) = p_{\theta_1}(X|Y, Z)p_{\theta_2}(\tilde{Y}|Y, X). \tag{3}$$

To infer latent variables $Z$ and $Y$ with only observable variables $X$ and $\tilde{Y}$, we design an inference network which model the variational distribution $q_\phi(Z, Y|\tilde{Y}, X)$. Specifically, let $q_{\phi_2}(Z|Y, X)$ and $q_{\phi_1}(Y|\tilde{Y}, X)$ be the distributions parameterized by learnable parameters $\phi_1$ and $\phi_2$, and then the posterior distribution can be decomposed as follows:

$$q_\phi(Z, Y|\tilde{Y}, X) = q_{\phi_2}(Z|Y, X)q_{\phi_1}(Y|\tilde{Y}, X), \tag{4}$$

where we do not include $\tilde{Y}$ as a conditioning variable in $q_{\phi_2}(Z|Y, X)$ because the causal graph implies $Z \perp\!\!\!\perp \tilde{Y}|X, Y$. One problem with this posterior form is that we cannot directly employ $q_{\phi_1}(Y|\tilde{Y}, X)$ to predict labels on the test data, on which $\tilde{Y}$ is absent.

To reduce computational costs and to allow our method efficiently infer clean labels, we approximate $q_{\phi_1}(Y|\tilde{Y}, X)$ by assuming that given the instance $X$, the clean label $Y$ is conditionally independent from the noisy label $\tilde{Y}$, i.e., $q_{\phi_1}(Y|\tilde{Y}, X) = q_{\phi_1}(Y|X)$. This approximation is expected not to have very large approximation error because the images contain sufficient information to predict the clean labels. Thus, we could simplify Eq. 4 as follows

$$q_\phi(Z, Y|X) = q_{\phi_2}(Z|Y, X)q_{\phi_1}(Y|X), \tag{5}$$

such that our encoder networks model $q_{\phi_2}(Z|Y, X)$ and $q_{\phi_1}(Y|X)$, respectively. This way, $q_{\phi_1}(Y|X)$ can be used to infer clean labels efficiently.[2] We also found that the encoder network modelling $q_{\phi_1}(Y|X)$ can directly act as a regularizer, which helps to identify $p_{\theta_2}(\tilde{Y}|Y, X)$. Moreover, be benefited from this, our method can serve as a general framework which can be easily integrated with the current discriminative label-noise methods [28, 16, 10], and we will showcase it by collaborating co-teaching [10] with our method.

**Optimization of Parameters** Because the marginal distribution $p_\theta(X, \tilde{Y})$ is usually intractable, to learn the set of parameters $\{\theta_1, \theta_2, \phi_1, \phi_2\}$ given only noisy data, we follow the variational inference framework [5] to maximize the negative evidence lower-bound $\text{ELBO}(x, \tilde{y})$ of the marginal likelihood of each datapoint $(x, \tilde{y})$ instead of maximizing the marginal likelihood itself. By ensembling our decoder and encoder networks, $\text{ELBO}(x, \tilde{y})$ is derived as follows:

$$\begin{aligned}\text{ELBO}(x, \tilde{y}) = {}&\mathbb{E}_{(z,y)\sim q_\phi(Z,Y|x)}\left[\log p_{\theta_1}(x|y, z)\right] + \mathbb{E}_{y\sim q_{\phi_1}(Y|x)}\left[\log p_{\theta_2}(\tilde{y}|y, x)\right] \\ &- kl(q_{\phi_1}(Y|x)\|p(Y)) - \mathbb{E}_{y\sim q_{\phi_1}(Y|x)}\left[kl(q_\phi(Z|y, x)\|p(Z))\right],\end{aligned} \tag{6}$$

where $kl(\cdot)$ is the Kullback–Leibler divergence between two distributions. The derivation details are left out in Appendix A. Our model learns the class-conditional distribution $P(X|Y)$ by maximizing the first expectation in ELBO, which is equivalent to minimizing the reconstruction loss [12]. By learning $P(X)$, the inference network $q_{\phi_1}(Y|X)$ has to select a suitable parameter $\phi^*$ which samples the $y$ and $z$ to minimize the reconstruction loss $\mathbb{E}_{(z,y)\sim q_\phi(Z,Y|x)}\left[\log p_{\theta_1}(x|y, z)\right]$. When the dimension of $Z$ is chosen to be much smaller than that of $X$, to obtain a smaller reconstruction error, the decoder has to utilize the information provided by $Y$, and force the value of $Y$ to be useful for

---

[2]Theoretically, this approximation is not necessary and can be easily removed. Specifically, we could let an encoder model $q_{\phi_1}(Y|\tilde{Y}, X)$. Then $q(Y|X)$ is obtained as $q(Y|X) = \sum_i q_{\phi_1}(Y|\tilde{Y} = i, X)q_{\phi_3}(\tilde{Y} = i|X)$, where $q_{\phi_3}(\tilde{Y}|X)$ is the noisy class-posterior distribution and can be learned by using noisy training data.

---

**Algorithm 1** CausalNL

---

**Input:** A noisy sample $S$, Average noise rate $\rho$, Total epoch $T_{max}$, Batch size $N$ .

1: **For** T $= 1, \ldots, T_{max}$:
2:   **For** mini-batch $\bar{S} = \{x_i\}_{i=0}^N, \tilde{L} = \{\tilde{y}_i\}_{i=0}^N$ in $S$:
3:     Feed $\bar{S}$ to encoders $\hat{q}_{\phi_1^1}$ and $\hat{q}_{\phi_1^2}$ to get clean label sets $L_1$ and $L_2$, respectively;
4:     Feed $(\bar{S}, L_1)$ to encoder $\hat{q}_{\phi_2^1}$ to get a representation set $H_1$, feed $(\bar{S}, L_2)$ to $\hat{q}_{\phi_2^2}$ to get $H_2$;
5:     Update $\hat{q}_{\phi_2^1}$ and $\hat{q}_{\phi_2^2}$ with co-teaching loss;
6:     Feed $(L_1, H_1)$ to decoder $\hat{p}_{\theta_1^1}$ to get reconstructed dataset $\bar{S}_1$, feed $(L_2, H_2)$ to $\hat{p}_{\theta_1^2}$ to get $\bar{S}_2$;
7:     Feed $(\bar{S}_1, L_1)$ to decoder $\hat{p}_{\theta_2^1}$ to get predicted noisy labels $\tilde{L}_1$, feed $(\bar{S}_2, L_2)$ to $\hat{p}_{\theta_2^2}$ to get $\tilde{L}_2$;
8:     Update networks $\hat{q}_{\phi_1^1}$, $\hat{q}_{\phi_2^1}$, $\hat{p}_{\theta_1^1}$ and $\hat{p}_{\theta_2^1}$ by calculating ELBO on $(\bar{S}, \bar{S}_1, \tilde{L}, \tilde{L}_1)$, update networks $\hat{q}_{\phi_1^2}, \hat{q}_{\phi_2^2}, \hat{p}_{\theta_1^2}$ and $\hat{p}_{\theta_2^2}$ by calculating ELBO on $(\bar{S}, \bar{S}_2, \tilde{L}, \tilde{L}_2)$;

    **Output:** The inference network $\hat{q}_{\phi_1^1}$.

---

prediction. Furthermore, we constrain the $Y$ to be a one-hot vector, and then $Y$ could be a cluster ID of which the manifold of the $X$ belongs.

So far, the latent variable $Y$ can be inferred as a cluster ID instead of a clean class ID. To further link the clusters to clean labels, a naive approach is to select some reliable examples and keep the cluster numbers to be consistent with the noisy labels on these examples. In such a way, the latent representation $Z$ and clean label $Y$ can be effectively inferred, and therefore, it encourages the identifiability of the transition relationship $p_{\theta_2}(\tilde{Y}|Y, X)$. To achieve this, instead of explicitly selecting the reliable example in advance, our method is trained end-to-end, i.e., reliable examples are selected dynamically during the update of parameters of our model by using the co-teaching technique [10]. The advantage of this approach is that the selection bias of the reliable example [6] can be greatly reduced. Intuitively, the accurately selected reliable examples can encourage the identifiability of $p_{\theta_2}(\tilde{Y}|Y, X)$ and $p_{\theta_1}(X|Y, Z)$, and the accurately estimated $p_{\theta_2}(\tilde{Y}|Y, X)$ and $p_{\theta_1}(X|Y, Z)$ will encourage the network to select more reliable examples.

### 3.2 Practical Implementation

Our method is summarized in Algorithm 1 and illustrated in Fig. 3. Here we introduce the structure of our model and loss functions.

**Model Structure** Because we incorporate co-teaching in our model training, we need to add a copy of the decoder and encoders in our method. As the two branches share the same architectures, we first present the details of the first branch and then briefly introduce the second branch.

For the first branch, we need a set of encoders and decoders to model the distributions in Eq. 3 and 5. Specifically, we have two encoder networks

$$Y_1 = \hat{q}_{\phi_1^1}(X), \ \ Z_1 \sim \hat{q}_{\phi_2^1}(X, Y_1)$$

for Eq. 5 and two decoder networks

$$X_1 = \hat{p}_{\theta_1^1}(Y_1, Z_1), \ \ \tilde{Y}_1 = \hat{p}_{\theta_2^1}(X_1, Y_1)$$

for Eq. 3. The first encoder $\hat{q}_{\phi_1^1}(X)$ takes an instance $X$ as input $\hat{q}_{\phi_1^1}(X)$ and output a predicted clean label $Y_1$. The second encoder $\hat{q}_{\phi_2^1}(X, Y_1)$ takes both the instance $X$ and the generated label $Y_1$ as input and outputs a latent feature $Z_1$. Then the generated $Y_1$ and $Z_1$ are passed through the decoder $\hat{p}_{\theta_1^1}(Y_1, Z_1)$ which will generate a reconstructed image $X_1$. Finally, the generated $X_1$ and $Y_1$ are the input to another decoder $\hat{p}_{\theta_2^1}(X_1, Y_1)$ which returns predicted noisy labels $\tilde{Y}_1$. It is worth mentioning that the reparameterization trick [12] is used for sampling, so as to allow backpropagation in $\hat{q}_{\phi_2^1}(X, Y_1)$.

Similarly, the encoder and decoder networks in the second branch are defined as follows:

$$Y_2 = \hat{q}_{\phi_1^2}(X), \ \ Z_2 \sim \hat{q}_{\phi_2^2}(X, Y_2), \qquad X_2 = \hat{p}_{\theta_1^2}(Y_2, Z_2), \ \ \tilde{Y}_2 = \hat{p}_{\theta_2^2}(X_2, Y_2).$$

During training, we let two encoders $\hat{q}_{\phi_1^1}(X)$ and $\hat{q}_{\phi_1^2}(X)$ teach each other given every mini-batch.

**Loss Functions** We divide the loss functions into two parts. The first part is the negative ELBO in Eq. 6, and the second part is a co-teaching loss. The detailed formulation is leaved in Appendix B.

For the negative ELBO, the first term $-\mathbb{E}_{(z,y)\sim q_\phi(Z,Y|x)}\left[\log p_{\theta_1}(x|y,z)\right]$ is a reconstruction loss, and we use the $\ell 1$ loss for reconstruction. The second term is $-\mathbb{E}_{y\sim q_{\phi_1}(Y|x)}\left[\log p_{\theta_2}(\tilde{y}|y,x)\right]$, which aims to learn noisy labels given inference $y$ and $x$, and can be simply replaced by the cross-entropy loss on outputs of both decoders $\hat{p}_{\theta_2^1}(X_1, Y_1)$ and $\hat{p}_{\theta_2^2}(X_2, Y_2)$ with the noisy labels contained in the training data. The additional two terms are two regularizers. To calculate $kl(q_{\phi_1}(Y|x)\|p(Y))$, we assume that the prior $P(Y)$ is a uniform distribution. Then minimizing $kl(q_{\phi_1}(Y|x)\|p(Y))$ is equivalent to maximizing the entropy of $q_{\phi_1}(Y|x)$ for each instance $x$, i.e., $-\sum_y q_{\phi_1}(y|x)\log q_{\phi_1}(y|x)$. The benefit for having this term is that it could reduce the overfiting problem of the inference network. For $\mathbb{E}_{y\sim q_{\phi_1}(Y|x)}\left[kl(q_\phi(Z|y,x)\|p(Z))\right]$, we let $p(Z)$ be a standard multivariate Gaussian distribution. Empirically, $q_\phi(Z|y,x)$ is modeled by the encoders $\hat{q}_{\phi_1^1}(X)$ and $\hat{q}_{\phi_1^2}(X)$ which are designed to be deterministic mappings, therefore, the expectation can be removed, and only the $kl$ term $kl(q_\phi(Z|y,x)\|p(Z))$ is left. When $p(Z)$ is a Gaussian distribution, the $kl$ term nicely has a closed form solution [12], i.e., $-\frac{1}{2}\sum_{j=1}^{J}(1+\log((\sigma_j)^2)-(\mu_j)^2-(\sigma_j)^2)$, where $J$ is the dimension of a latent representation $z$, and $\sigma_j$ and $\mu_j$ are the encoder outputs.

For the co-teaching loss, we follow the work of Han et al. [10]. Intuitively, two encoders $\hat{q}_{\phi_1^1}(X)$ and $\hat{q}_{\phi_1^2}(X)$ feed all data forward and selects some data of possibly clean labels. Then, two networks communicate with each other to select possible clean data in this mini-batch and use them for training. Finally, each encoder backpropagates over the data selected by its peer network and updates itself by cross-entropy loss.

## 4  Experiments

In this section, we compare the classification accuracy of proposed method with popular label-noise learning algorithms [15, 19, 11, 10, 28, 36, 16] on both synthetic and real-world datasets.

### 4.1  Experimental Setup

**Datasets** We examine the efficacy of our approach on manually corrupted versions of four datasets, i.e., *FashionMNIST* [31], *SVHN* [18], *CIFAR10*, *CIFAR100* [13], and one real-world noisy dataset, i.e., *Clothing1M* [32]. *FashionMNIST* contains 60,000 training images and 10,000 test images with 10 classes; *SVHN* contains 73,257 training images and 26,032 test images with 10 classes. *CIFAR10* contains 50,000 training images and 10,000 test images. *CIFAR10* and *CIFAR100* both contain 50,000 training images and 10,000 test images but the former has 10 classes of images, and the latter has 10 classes of images. The four datasets contain clean data. We add instance-dependent label noise to the training sets manually according to Xia et al. [30]. *Clothing1M* has 1M images with real-world noisy labels and 10k images with clean labels for testing. For all the synthetic noisy datasets, the experiments are repeated 5 times.

**Network structure and optimization** For a fair comparison, all experiments are conducted on NVIDIA Tesla V100, and all methods are implemented by PyTorch. Dimension of the latent representation $Z$ is set to 25 for all synthetic noisy datasets. For encoder networks $\hat{q}_{\phi_1^1}(X)$ and $\hat{q}_{\phi_1^2}(X)$, we use the same network structures with the baseline method. Specially, we use a ResNet-18 network for *FashionMNIST*, a ResNet-34 network for *SVHN* and *CIFAR10*, a ResNet-50 network for *CIFAR100* without pretraining. For *Clothing1M*, we use ResNet-50 networks pre-trained on ImageNet. The data-augmentation methods random crop and horizontal flip are used for our method. For Clothing1M, we use a ResNet-50 network pre-trained on ImageNet, and the clean training data is not used. The dimensionality of the latent representation $Z$ is set to 100. Due to limited space, we leave the detailed structure of other decoders and encoders in Appendix C.

**Baselines and measurements** We compare the proposed method with the following state-of-the-art approaches: (i). CE, which trains the standard deep network with the cross-entropy loss on noisy datasets. (ii). Decoupling [16], which trains two networks on samples whose predictions from the two networks are different. (iii). MentorNet [11] and Co-teaching [10], which mainly handle noisy labels

Table 1: Means and standard deviations (percentage) of classification accuracy on *FashionMNIST* with different label noise levels.

|  | IDN-20% | IDN-30% | IDN-40% | IDN-45% | IDN-50% |
|---|---|---|---|---|---|
| CE | 88.54±0.32 | 88.38±0.42 | 84.22±0.35 | 69.72±0.72 | 52.32±0.68 |
| Co-teaching | 91.21±0.31 | 90.30±0.42 | 89.10±0.29 | 86.78±0.90 | 63.22±1.56 |
| Decoupling | 90.70±0.28 | 90.34±0.36 | 88.78±0.44 | 87.54±0.53 | 68.32±1.77 |
| MentorNet | 91.57±0.29 | 90.52±0.41 | 88.14±0.76 | 85.12±0.76 | 61.62±1.42 |
| Mixup | 88.68±0.37 | 88.02±0.37 | 85.47±0.55 | 79.57±0.75 | 66.02±2.58 |
| Forward | 90.05±0.43 | 88.65±0.43 | 86.27±0.48 | 73.35±1.03 | 58.23±3.14 |
| Reweight | 90.27±0.27 | 89.58±0.37 | 87.04±0.32 | 80.69±0.89 | 64.13±1.23 |
| T-Revision | **91.58**±0.31 | 90.11±0.61 | 89.46±0.42 | 84.01±1.14 | 68.99±1.04 |
| CausalNL | 90.84±0.31 | **90.68**±0.37 | **90.01**±0.45 | **88.75**±0.81 | **78.19**±1.01 |

Table 2: Means and standard deviations (percentage) of classification accuracy on *SVHN* with different label noise levels.

|  | IDN-20% | IDN-30% | IDN-40% | IDN-45% | IDN-50% |
|---|---|---|---|---|---|
| CE | 91.51±0.45 | 91.21±0.43 | 87.87±1.12 | 67.15±1.65 | 51.01±3.62 |
| Co-teaching | 93.93±0.31 | 92.06±0.31 | 91.93±0.81 | 89.33±0.71 | 67.62±1.99 |
| Decoupling | 90.02±0.25 | 91.59±0.25 | 88.27±0.42 | 84.57±0.89 | 65.14±2.79 |
| MentorNet | **94.08**±0.12 | 92.73±0.37 | 90.41±0.49 | 87.45±0.75 | 61.23±2.82 |
| Mixup | 89.73±0.37 | 90.02±0.35 | 85.47±0.63 | 82.41±0.62 | 68.95±2.58 |
| Forward | 91.89±0.31 | 91.59±0.23 | 89.33±0.53 | 80.15±1.91 | 62.53±3.35 |
| Reweight | 92.44±0.34 | 92.32±0.51 | 91.31±0.67 | 85.93±0.84 | 64.13±3.75 |
| T-Revision | 93.14±0.53 | 93.51±0.74 | 92.65±0.76 | 88.54±1.58 | 64.51±3.42 |
| CausalNL | 94.06±0.23 | **93.86**±0.65 | **93.82**±0.64 | **93.19**±0.93 | **85.41**±2.95 |

Table 3: Means and standard deviations (percentage) of classification accuracy on *CIFAR10* with different label noise levels.

|  | IDN-20% | IDN-30% | IDN-40% | IDN-45% | IDN-50% |
|---|---|---|---|---|---|
| CE | 75.81±0.26 | 69.15±0.65 | 62.45±0.86 | 51.72±1.34 | 39.42±2.52 |
| Co-teaching | 80.96±0.31 | 78.56±0.61 | 73.41±0.78 | 71.60±0.79 | 45.92±2.21 |
| Decoupling | 78.71±0.15 | 75.17±0.58 | 61.73±0.34 | 58.61±1.73 | 50.43±2.19 |
| MentorNet | 81.03±0.24 | 77.22±0.47 | 71.83±0.49 | 66.18±0.64 | 47.89±2.03 |
| Mixup | 73.17±0.34 | 70.02±0.31 | 61.56±0.71 | 56.45±0.67 | 48.95±2.58 |
| Forward | 74.64±0.26 | 69.75±0.56 | 60.21±0.75 | 48.81±2.59 | 46.27±1.30 |
| Reweight | 76.23±0.25 | 70.12±0.72 | 62.58±0.46 | 51.54±0.92 | 45.46±2.56 |
| T-Revision | 76.15±0.37 | 70.36±0.54 | 64.09±0.37 | 52.42±1.01 | 49.02±2.13 |
| CausalNL | **81.47**±0.32 | **80.38**±0.44 | **77.53**±0.45 | **78.60**±1.06 | **67.39**±1.24 |

by training on instances with small loss values. (iv). Forward [19], Reweight [15], and T-Revision [28]. These approaches utilize a class-dependent transition matrix $T$ to correct the loss function. For these baselines, we follow the experiments settings of the original papers. We report average test accuracy on over the last ten epochs of each model on the clean test set. Higher classification accuracy means that the algorithm is more robust to the label noise.

## 4.2 Classification Accuracy Evaluation

**Results on synthetic noisy datasets**  Tables 1, 2, 3, and 4 report the classification accuracy on the datasets of *F-MNIST*, *SVHN*, *CIFAR-10*, and *CIFAR100*, respectively. The synthetic experiments reveal that our method is powerful in handling instance-dependent label noise particularly in the situation of high noise rates. Specifically, for all datasets, the classification accuracy of our method

Table 4: Means and standard deviations (percentage) of classification accuracy on *CIFAR100* with different label noise levels.

|  | IDN-20% | IDN-30% | IDN-40% | IDN-45% | IDN-50% |
|---|---|---|---|---|---|
| CE | 30.42±0.44 | 24.15±0.78 | 21.45±0.70 | 15.23±1.32 | 14.42±2.21 |
| Co-teaching | 37.96±0.53 | 33.43±0.74 | 28.04±1.43 | 25.60±0.93 | 23.97±1.91 |
| Decoupling | 36.53±0.49 | 30.93±0.88 | 27.85±0.91 | 23.81±1.31 | 19.59±2.12 |
| MentorNet | 38.91±0.54 | 34.23±0.73 | 31.89±1.19 | 27.53±1.23 | 24.15±2.31 |
| Mixup | 32.92±0.76 | 29.76±0.87 | 25.92±1.26 | 23.13±2.15 | 21.31±1.32 |
| Forward | 36.38±0.92 | 33.17±0.73 | 26.75±0.93 | 21.93±1.29 | 19.27±2.11 |
| Reweight | 36.73±0.72 | 31.91±0.91 | 28.39±1.46 | 24.12±1.41 | 20.23±1.23 |
| T-Revision | 37.24±0.85 | 36.54±0.79 | 27.23±1.13 | 25.53±1.94 | 22.54±1.95 |
| CausalNL | **41.47**±0.43 | **40.98**±0.62 | **34.02**±0.95 | **33.34**±1.13 | **32.129**±2.23 |

Table 5: Classification accuracy on *Clothing1M*. In the experiments, only noisy samples are exploited to train and validate the deep model.

| CE | Decoupling | MentorNet | Co-teaching | Forward | Reweight | T-Revision | caualNL |
|---|---|---|---|---|---|---|---|
| 68.88 | 54.53 | 56.79 | 60.15 | 69.91 | 70.40 | 70.97 | **72.24** |

decrease much slower than that of baseline methods. Additionally, the classification accuracies on these datasets are improved by using CausalNL, which implies that our method should capture the underlying data generation process, and then $Y$ should be a cause of $X$ for all these datasets.

For noisy *F-MNIST*, *SVHN* and *CIFAR-10*, in the easy case IDN-20%, almost all methods work well. When the noise rate is 30%, the advantages of causalNL begin to show. We surpassed all methods obviously. When the noise rate raises, all the baselines are gradually defeated. Finally, in the hardest case, i.e., IDN-50%, the superiority of causalNL widens the gap of performance. The classification accuracy of causalNL is at least over 10% higher than the best baseline method. For noisy *CIFAR-100*, none of the methods works well. However, causalNL still overtakes the other methods with clear gaps for all different levels of noise rate.

**Results on the real-world noisy dataset** On the real-world noisy dataset *Clothing1M*, our method causalNL outperforms all the baselines, as shown in Table 5. The experimental results also show that the noise type in *Clothing1M* is more likely to be instance-dependent label noise, suggesting that the instance-independent assumption on the transition matrix sometimes can be strong.

## 5 Conclusion

In this paper, we have investigated how to use $P(X)$ to help learn instance-dependent label noise. Specifically, previous assumptions are made on the transition matrix, and the assumptions are hard to be verified and might be violated on real-world datasets. From inspired by a causal perspective, when $Y$ is a cause of $X$, then $P(X)$ should contain useful information to infer the clean label $Y$. We propose a novel generative approach called causalNL for instance-dependent label-noise learning. Our model makes use of the causal graph to contribute to the identifiability of the transition matrix, and therefore helps learn clean labels. In order to learn $P(X)$, compared to the previous methods, our method contains more parameters. But experiments on both synthetic and real-world noisy datasets show that a bit sacrifice on computational efficiency is worth it, i.e., the classification accuracy of casualNL significantly outperforms all the state-of-the-art methods. Additionally, the results also indicates that in classification problems, $Y$ can usually be considered as a cause of $X$, and suggests that the understanding and modeling of the data generation process can help leverage additional information that is useful in solving advanced machine learning problems concerning the relationship between different modules of the data joint distribution. In our future work, we will study the theoretical properties of our method and establish identifiability results under certain assumptions on the data-generative process.

## Acknowledgments

TL was partially supported by Australian Research Council Projects DP-180103424, DE-190101473, and IC-190100031. GM was supported by Australian Research Council Project DE210101624. BH was supported by supported by the RGC Early Career Scheme No. 22200720 and NSFC Young Scientists Fund No. 62006202. GN was supported by JST AIP Acceleration Research Grant Number JPMJCR20U3, Japan. KZ was supported in part by the National Institutes of Health (NIH) under Contract R01HL159805, by the United States Air Force under Contract No. FA8650-17-C7715, by the NSF-Convergence Accelerator Track-D award #2134901, and by a grant from Apple. The NIH or NSF is not responsible for the views reported in this article. The authors thank the reviewers and the meta-reviewer for their helpful and constructive comments on this work.

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
