# Supplementary to "Instance-dependent Label-noise Learning under a Structural Causal Model"

**Yu Yao**[1]    **Tongliang Liu**[1][†]   **Mingming Gong**[2]
**Bo Han**[3]     **Gang Niu**[4]     **Kun Zhang**[5]
[1]TML Lab, University of Sydney; [2]University of Melbourne; [3]Hong Kong Baptist University;
[4]RIKEN AIP; [5]Carnegie Mellon University

## Appendix

### Appendix A: Derivation Details of evidence lower-bound (ELBO)

In this section, we show the derivation details of $\text{ELBO}(x, \tilde{y})$.
Recall that the causal decomposition of the instance-dependent label noise is

$$P(X, \tilde{Y}, Y, Z) = P(Y)P(Z)P(X|Y, Z)P(\tilde{Y}|Y, X).$$

Our encoders model following distributions

$$q_\phi(Z, Y|X) = q_{\phi_2}(Z|Y, X)q_{\phi_1}(Y|X),$$

and decoders model the following distributions

$$p_\theta(X, \tilde{Y}|Y, Z) = p_{\theta_1}(X|Y, Z)p_{\theta_2}(\tilde{Y}|Y, X).$$

Now, we start with maximizing the log-likelihood $p_\theta(x, \tilde{y})$ of each datapoint $(x, \tilde{y})$.

$$
\begin{aligned}
\log p_\theta(x, \tilde{y}) &= \log \int_z \int_y p_\theta(x, \tilde{y}, z, y)\mathrm{d}y\mathrm{d}z \\
&= \log \int_z \int_y p_\theta(x, \tilde{y}, z, y)\frac{q_\phi(z, y|x)}{q_\phi(z, y|x)}\mathrm{d}y\mathrm{d}z \\
&= \log \mathbb{E}_{(z,y)\sim q_\phi(Z,Y|x)}\left[\frac{p_\theta(x, \tilde{y}, z, y)}{q_\phi(z, y|x)}\right] \\
&\geq \mathbb{E}_{(z,y)\sim q_\phi(Z,Y|x)}\left[\log \frac{p_\theta(x, \tilde{y}, z, y)}{q_\phi(z, y|x)}\right] := \text{ELBO}(x, \tilde{y}) \\
&= \mathbb{E}_{(z,y)\sim q_\phi(Z,Y|x)}\left[\log \frac{p(z)p(y)p_{\theta_1}(x|y, z)p_{\theta_2}(\tilde{y}|y, x))}{q_\phi(z, y|x)}\right] \\
&= \mathbb{E}_{(z,y)\sim q_\phi(Z,Y|x)}[\log\left(p_{\theta_1}(x|y, z)\right)] + \mathbb{E}_{(z,y)\sim q_\phi(Z,Y|x)}[\log\left(p_{\theta_2}(\tilde{y}|y, x)\right)] \\
&\quad + \mathbb{E}_{(z,y)\sim q_\phi(Z,Y|x)}\left[\log\left(\frac{p(z)p(y)}{q_{\phi_2}(z|y, x)q_{\phi_1}(y|x)}\right)\right]
\end{aligned}
\tag{1}
$$

The $\text{ELBO}(x, \tilde{y})$ above can be further simplified. Specifically,

$$
\begin{aligned}
\mathbb{E}_{(z,y)\sim q_\phi(Z,Y|x)}[\log\left(p_{\theta_2}(\tilde{y}|y, x)\right)] &= \mathbb{E}_{y\sim q_{\phi_1}(Y|x)}\mathbb{E}_{z\sim q_{\phi_2}(Z|y, x)}[\log\left(p_{\theta_2}(\tilde{y}|y, x)\right)] \\
&= \mathbb{E}_{y\sim q_{\phi_1}(Y|x)}[\log\left(p_{\theta_2}(\tilde{y}|y, x)\right)],
\end{aligned}
\tag{2}
$$

---

[†]Correspondence to Tongliang Liu (tongliang.liu@sydney.edu.au).

35th Conference on Neural Information Processing Systems (NeurIPS 2021).

and similarly,

$$\mathbb{E}_{(z,y)\sim q_\phi(Z,Y|x)}\left[\log\left(\frac{p(z)p(y)}{q_{\phi_2}(z|y,x)q_{\phi_1}(y|x)}\right)\right]$$

$$=\mathbb{E}_{y\sim q_{\phi_1}(Y|x)}\mathbb{E}_{z\sim q_{\phi_2}(Z|y,x)}\left[\log\left(\frac{p(z)p(y)}{q_{\phi_2}(z|y,x)q_{\phi_1}(y|x)}\right)\right]$$

$$=\mathbb{E}_{y\sim q_{\phi_1}(Y|x)}\mathbb{E}_{z\sim q_{\phi_2}(Z|y,x)}\left[\log\left(\frac{p(y)}{q_{\phi_1}(y|x)}\right)\right]+\mathbb{E}_{y\sim q_{\phi_1}(Y|x)}\mathbb{E}_{z\sim q_{\phi_2}(Z|y,x)}\left[\log\left(\frac{p(z)}{q_{\phi_2}(z|y,x)}\right)\right]$$

$$=\mathbb{E}_{y\sim q_{\phi_1}(Y|x)}\left[\log\left(\frac{p(y)}{q_{\phi_1}(y|x)}\right)\right]+\mathbb{E}_{y\sim q_{\phi_1}(Y|x)}\mathbb{E}_{z\sim q_{\phi_2}(Z|y,x)}\left[\log\left(\frac{p(z)}{q_{\phi_2}(z|y,x)}\right)\right]$$

$$=-kl(q_{\phi_1}(Y|x)\|p(Y))-\mathbb{E}_{y\sim q_{\phi_1}(Y|x)}\left[kl(q_\phi(Z|y,x)\|p(Z))\right], \tag{3}$$

By combing Eq. 1, Eq. 2 and Eq. 3, we get

$$\mathrm{ELBO}(x,\tilde{y})=\mathbb{E}_{(z,y)\sim q_\phi(Z,Y|x)}\left[\log p_{\theta_1}(x|y,z)\right]+\mathbb{E}_{y\sim q_{\phi_1}(Y|x)}\left[\log p_{\theta_2}(\tilde{y}|y,x)\right]$$
$$-kl(q_{\phi_1}(Y|x)\|p(Y))-\mathbb{E}_{y\sim q_{\phi_1}(Y|x)}\left[kl(q_\phi(Z|y,x)\|p(Z))\right],$$

which is the ELBO in our main paper.

## Appendix B: Loss Functions

In this section, we provide the empirical solution of the ELBO and co-teaching loss. Remind that our encoder networks and decoder networks in the the first branch are defined as follows

$$Y_1=\hat{q}_{\phi_1^1}(X),\;\;Z_1\sim\hat{q}_{\phi_2^1}(X,Y_1),\;\;\;\;\;\;\;\;X_1=\hat{p}_{\theta_1^1}(Y_1,Z_1),\;\;\tilde{Y}_1=\hat{p}_{\theta_2^1}(X_1,Y_1),$$

Let $S$ be the noisy training set, and $d^2$ be the dimension of an instance $x$. Let $y_1$ and $z_1$ be the estimated clean label and latent representation for the instance $x$, respectively, by the first branch. As mentioned in our main paper (see Section 3.2), the negative ELBO loss is to minimize 1). a reconstruction loss between each instance $x$ and $\hat{p}_{\theta_1^1}(x,y_1)$; 2). a cross-entropy loss between noisy labels $\hat{p}_{\theta_2^1}(x_1,x_1)$ and $\tilde{y}$; 3). a cross-entropy loss between $\hat{q}_{\phi_2^1}(x,y_1)$ and uniform distribution $P(Y)$; 4). a cross-entropy loss between $\hat{q}_{\phi_2^1}(x,y_1)$ and Gaussian distribution $P(Z)$. Specifically, the empirical version of the ELBO for the first branch is as follows.

$$\sum_{(x,\tilde{y})\in S}\hat{\mathrm{ELBO}}^1(x,\tilde{y})=\sum_{(x,\tilde{y})\in S}\left[\beta_0\frac{1}{d^2}\|x-\hat{p}_{\theta_1^1}(y_1,z_1)\|_1-\beta_1\tilde{y}\log\hat{p}_{\theta_2^1}(x_1,y_1)\right.$$
$$\left.+\beta_2\hat{q}_{\phi_1^1}(x)\log\hat{q}_{\phi_1^1}(x)+\beta_3\sum_{j=1}^J(1+\log((\sigma_j)^2)-(\mu_j)^2-(\sigma_j)^2)\right].$$

The hyper-parameter $\beta_0$ and $\beta_1$ are set to 0.1, and $\beta_2$ are set to $1e-5$ because encouraging the distribution to be uniform on a small min-batch (i.e., 128) could have a large estimation error. The hyper-parameter $\beta_3$ are set to 0.01. The empirical version of the ELBO for the second branch shares the same settings as the first branch.

For co-teaching loss, we directly follow Han et al. [1]. Intuitively, in each mini-batch data, both encoders $\hat{q}_{\phi_1^1}(X)$ and $\hat{q}_{\phi_1^2}(X)$ select their small-loss instances as the useful knowledge and exchange the knowledge to each other by a cross-entropy loss. The number of the small-loss instances used for training decays with respect to the training epoch. The experimental settings for co-teaching loss are the same as the settings in the original paper [1].

## Appendix C: More Experimental Settings

In this section, we summarize the network structures for different datasets. The network structure for modeling $q_{\phi_1}(Y|X)$ and the dimension of the latent representation $Z$ has been discussed in our main paper. For the optimization method, we use Adam with the default learning rate $1e-3$ in Pytorch. The source code has been included in our supplementary material.

For *FashionMNIST* [3], *SVHN* [2], *CIFAR10* and *CIFAR100*, we use the same number of hidden layers and feature maps. Specifically, 1). we model $q_{\phi_2}(Z|Y, X)$ and $p_{\theta_2}(\tilde{Y}|Y, X)$ by two 4-hidden-layer convolutional networks, and the corresponding feature maps are 32, 64, 128 and 256; 2). we model $p_{\theta_1}(X|Y, Z)$ by a 4-hidden-layer transposed-convolutional network, and the corresponding feature maps are 256, 128, 64 and 32. We ran 150 epochs for each experiment on these datasets.

For *Clothing1M* [4], 1). we model $q_{\phi_2}(Z|Y, X)$ and $p_{\theta_2}(\tilde{Y}|Y, X)$ by two 5-hidden-layer convolutional networks, and the corresponding feature maps are 32, 64, 128, 256, 512; 2). we model $p_{\theta_1}(X|Y, Z)$ by a 5-hidden-layer transposed-convolutional network, and the corresponding feature maps are 512, 256, 128, 64 and 32. We ran 40 epochs on *Clothing1M*.