# OpenReview forum: "Instance-dependent Label-noise Learning under a Structural Causal Model"
_NeurIPS.cc/2021/Conference — NeurIPS 2021 Poster_

### Official Review · Reviewer_LYgh · 2021-07-14

**Rating:** 7
**Confidence:** 5

**Summary:**

In this paper, the authors handle the label noise problem by exploiting the causal information, which is a new perspective. The setting of instance-dependent label noise learning is more general and realistic in real life. The motivation of this paper is strong. The proposed method also has a good empirical performance and can be easily integrated into current methods.

**Limitations And Societal Impact:**

1. If the noise is dependent on the instance, the proposed method works well, but will the proposed method introduce bias when the noise is independent of the instance? I think it is better to add some discussion on this point.
2. The experimental settings such as network structures and hyper-parameters of Figure 2 are not clearly introduced neither in the supplementary material or main paper. Please clarify it.
3. There might have a typo in either Table2 or Table 3. For IDN-40%, the accuracy of Mixup in Table 2 and Table 3 are exactly same, and the accuracy of Mixup in Table 3 is higher than the proposed method.

**Main Review:**

The authors have proposed a novel method for label-noise learning. Different from most of the current works, instead of making assumptions on the transition relationship, the authors exploit the dependence relationship among variables when generating the noisy data.  In such a way, the important information for helping identify the transition relationship and clean labels can be revealed. To learn the useful information, a structural generative model is proposed and outperforms the state-of-the-art method. This paper is well organized. The novelty of this paper is high. To be best of my knowledge, this is the first paper to handle the instance-dependent label-noise problem from a causal perspective. But some of my major concerns might have to be addressed.

**Time Spent Reviewing:**

5

---

> ### Author Response · Authors · 2021-08-09
> **Response to Reviewer LYgh by Paper5377 Authors**
>
> **Questions**
> _______________________________________________________________________________________________________________
> + If the noise is dependent on the instance, the proposed method works well, but will the proposed method introduce bias when the noise is independent of the instance? I think it is better to add some discussion on this point.
> + The experimental settings such as network structures and hyper-parameters of Figure 2 are not clearly introduced neither in the supplementary material or main paper. Please clarify it.
> + There might have a typo in either Table2 or Table 3. For IDN-40$\\%$, the accuracy of Mixup in Table 2 and Table 3 are exactly same, and the accuracy of Mixup in Table 3 is higher than the proposed method.
> _____________________________________________________________________________________________________________
> \
> \
> \
> **Q1.** If the noise is dependent on the instance, the proposed method works well, but will the proposed method introduce bias when the noise is independent of the instance?
>
> **A1.** Thank you for raising this insightful concern.  Our work will not introduce extra bias when the noise is independent of the instance, because the instance-independent label noise model is a special case of our causal graph in Fig. 1. Please kindly see detailed discussions in the response to A1 of Reviewer 38W1.
> \
> \
> \
> **Q2.** The experimental settings such as network structures and hyper-parameters of Figure 2 are not clearly introduced neither in the supplementary material or main paper. Please clarify it.
>
> **A2.** Thank you very much for pointing this out. For our method, the encoders and decoders do not have any hidden layer. The size of latent feature $Z$ is set to $2$. The min-batch size is set to $512$. The hyper-parameter $\beta_0$ and $\beta_1$ are set to $0.1$, $\beta_2$ are set to $0.00001$, $\beta_3$ are set to $0.01$. For coteaching and mixup, we use the similar network structure as the encoder $q_{\phi_1}(Y|X)$. We will add these details to the supplementary martial, in light of your comment.
> \
> \
> \
> **Q3.** There might have a typo in either Table2 or Table 3. For IDN-$40\\%$, the accuracy of Mixup in Table 2 and Table 3 are exactly same, and the accuracy of Mixup in Table 3 is higher than the proposed method.
>
> **A3.** Thank you very much for pointing this out. We sincerely apologize for the confusion. It is indeed a typo. For IDN-$40\\%$, the accuracy of Mixup in Table 3 should be $64.12\\%$. We will carefully go through the paper to correct some other typos as well.

---

### Official Review · Reviewer_yi5B · 2021-07-15

**Rating:** 7
**Confidence:** 4

**Summary:**

The paper describes an approach to utilize causal information in the form of a Structural Causal Model (SCM) to facilitate learning under instance-dependent label noise. The approach utilizes semi-supervised learning principles, relying on the causal origin of this effect due to the entanglement between the class conditional $P(Y|X)$ and the marginal $P(X)$ in the anti-causal direction.

**Limitations And Societal Impact:**

The author clearly addresses the assumptions made to make the noise transition model identifiable. These assumptions defining the limitations of the model.
On the societal impact: Given this method can handle instance dependent label-noise, can it be applied in a situation where this instance dependence introduces a bias? For example if a protected attribute of an individual influences the noisy labels.

**Main Review:**

The described approach utilizes the properties of anti-causal learning in a novel area by applying it to learning with noisy labels. It is applicable in the more complex case of instance-dependent label noise where most of the previous methods are not applicable, or require special assumptions.
Clearly the present method also require some assumptions, but these are mainly causal, therefore more natural for a human expert to specify. According to the experiments the method achieve consistent improvement over the state of the art, and this improvement is increasing with increasing label-noise.

My main question is about the assumed causal model:
If the noisy labels are generated using crowdsourcing, and the worker only see the image itself, what mechanism provide the link between $Y$ and $\tilde{Y}$ ? What type of side-channel relay this mechanism?
If the human expert providing noisy labels can see the real object, the model is clearly valid, but this is not the typical case.

What condition make the approximation that the clean label $Y$ is conditionally independent from the noisy label $\tilde{Y}$ given $X$ in the inference model acceptable, while the approach still benefits from assuming direct causal effect from $Y$ to $\tilde{Y}$? Or in other way, what would happen if we assume $X$ is enough to generate $\tilde{Y}$.

Algorithm 1 lists $\rho$ (the noise rate) as its input, where it is used? How realistic it is to know this value?

line 132 (minor error in text) but it is impossible [to tell] which class each cluster belongs to.


**Time Spent Reviewing:**

5

---

> ### Author Response · Authors · 2021-08-09
> **Response to Reviewer yi5B by Paper5377 Authors**
>
> **Questions**
> _____________________________________________________________________________________________________________
> + My main question is about the assumed causal model: If the noisy labels are generated using crowdsourc-ing, and the worker only see the image itself, what mechanism provide the link between $Y$ and $\tilde{Y}$ ? What type of side-channel relay this mechanism? If the human expert providing noisy labels can see the real object, the model is clearly valid, but this is not the typical case.
> + What condition make the approximation that the clean label Y is conditionally independent from the noisy label $\tilde{Y}$ given $X$ in the inference model acceptable, while the approach still benefits from assuming direct causal effect from $Y$ to $\tilde{Y}$? Or in other way, what would happen if we assume $X$ is enough to gen-erate $\tilde{Y}$.
> + Algorithm 1 lists $\rho$ (the noise rate) as its input, where it is used? How realistic it is to know this value?
> + line 132 (minor error in text) but it is impossible [to tell] which class each cluster belongs to.
>
> Limitations And Societal Impact:
>
> On the societal impact: Given this method can handle instance dependent label-noise, can it be applied in a situa-tion where this instance dependence introduces a bias? For example if a protected attribute of an individual influ-ences the noisy labels.
> _____________________________________________________________________________________________________________
> \
> \
> \
> **Q1.** If the noisy labels are generated using crowdsourcing, and the worker only sees the image itself, what mechanism provides the link between $Y$ and $\tilde{Y}$? What type of side-channel relay this mechanism? If the human expert providing noisy labels can see the real object, the model is clearly valid, but this is not the typical case.
>
> **A1.** Thank you for making this great point. We think that, in general, if the worker only sees the image itself, then it is hard to link $Y$ and $\tilde{Y}$. But if the set of clean labels is given, then it can provide a link between $Y$ and $\tilde{Y}$. For example, if we only collect images from an online alcohol shopping website, although we do not have the clean label for each instance, but the clean label set (such as {wine, vodka, whiskey, ..., xo}) is expected to be already on our mind. Then, when we annotate a (noisy) label for an image, and the noisy label is selected according to the clean label set. It is also worth mentioning that, in general, if there is no link between Y and $\tilde{Y}$, then one can even consider $Z$ as $Y$, that is, the link between $Y$ and $\tilde{Y}$ helps determine what underlying property of $X$ to estimate.
> \
> \
> \
> **Q2.** What condition make the approximation that the clean label $Y$ is conditionally independent from the noisy label $\tilde{Y}$? given $X$ in the inference model acceptable, while the approach still benefits from assuming direct causal effect from $Y$ to $\tilde{Y}$? Or in other way, what would happen if we assumed $X$ is enough to generate $\tilde{Y}$.
>
> **A2.** Thanks for the great question. We will add the following discussion into our main paper. The only purpose of using this approximation is to reduce the computational cost; this approximation is not necessary on the theory side, and it can be easily removed. Specifically, we could let an encoder model $q_{\phi_1}(Y|\tilde{Y}, X)$, and then $q(Y|X)$ is obtained as $q(Y|X) = \sum_i q_{\phi_1}(Y|\tilde{Y}=i, X) q_{\phi_3}(\tilde{Y}=i| X)$, where $q_{\phi_3}(\tilde{Y}| X)$ is the noisy class posterior and can be learned by using noisy training data. By using this approximation, the extra model $q_{\phi_3}(\tilde{Y}| X)$ is not required to be learned. Empirically, we found the classification accuracy has been significantly improved, and therefore the trade-off of using the approximation is acceptable. Additionally, this kind of approximation is commonly used in Bayesian inference (e.g., [3]).
> \
> \
> \
> **Q3.** Algorithm 1 lists $\rho$ (the noise rate) as its input, where it is used? How realistic it is to know this value?
>
> **A3.** The requirement of the noise rate comes from co-teaching. Specifically, it is used in the objective function of co-teaching to facilitate confidence example selection and regularize P(Y|X). The requirement of the noise rate can be removed if we change co-teaching into other suitable noise-learning methods such as decoupling [1] (this can be simply achieved by a modification of the co-teaching loss).
> In general, the noise rate may be unknown. The noise rate for instance-dependent label noise is unidentifiable without suitable assumptions. To identify it, we either need to access the clean labels for a small part of the training set or need to add assumptions on the noise type, e.g., that under the part-dependent assumption, the noise rate can be estimated [2].
> \
> \
> \
> **Q4.** line 132 (minor error in text) but it is impossible [to tell] which class each cluster belongs to.
>
> **A4.** Thank you very much and will be corrected.
> \
> \
> \
> **Q5 (Societal impact).** Given this method can handle instance-dependent label-noise, can it be applied in a situation where this instance dependence introduces a bias?
>
> **A5.** Thank you for this insightful question. Conceptually, our method might can be extended to overcome the bias problem by reconstructing a new causal graph that takes the effect of the protected attribute into consideration. For example, the protected attribute can be a cause of the noisy label. Handling the biased label-noise problem can have strong practical value, and this extension does not seem trivial. We believe that it can be an important line of future research.
> \
> \
> \
> **References**
> _____________________________________________________________________________________________________________
> [1]. Malach, Eran, and Shai Shalev-Shwartz. "Decoupling" when to update" from" how to update"." arXiv preprint arXiv:1706.02613 (2017).
>
> [2]. Xia, Xiaobo, et al. "Part-dependent label noise: Towards instance-dependent label noise." Advances in Neural Information Processing Systems 33 (2020).
>
> [3]. Zhang, Cheng, Kun Zhang, and Yingzhen Li. "A causal view on robustness of neural networks." arXiv preprint arXiv:2005.01095 (2020).

---

### Official Review · Reviewer_38W1 · 2021-07-18

**Rating:** 5
**Confidence:** 3

**Summary:**

The paper addresses the question of learning with noisy label
The idea is to consider an underlying causal graph explaining the observed noisy label.
To do so the authors consider that there is an underlying generative process and that the observed noisy label is  generated both by the true label and a relevant representation of the image of interest
The paper the provides a variational inference method based on this generative process and on a relevant representation of the noisy image
This is combined with coteaching already introduced in [9]
Experiments are thereafter performed on several datasets and results (taking into account std) are quite comparable with the T-Revision approach and outperforms co-teaching when the noise is severe


**Ethical Concerns:**

No concern

**Limitations And Societal Impact:**

No concern

**Main Review:**

* The idea of modeling noise generation through a generative process is original and new
* More precision should be given about the way some noise is added to the data. A reference is given but one may wonder the impact of the way the noise is added on the performance of the algorithm, especially when using variational inference. Can you add a discussion on the impact of the choice of the noise model?
* Can you detail how noise impact precision and recall for each model?
* What about the choice of prior on Z and Y?
* The claim is that the framework is general and that it can be combined with any method dealing with label noise. Can you explain the choice of co-teaching? What about combination of the framework with other label-noise learning methods?

**Time Spent Reviewing:**

2h

---

> ### Author Response · Authors · 2021-08-09
> **Response to Reviewer 38W1 by Paper5377 Authors**
>
> **Questions**
> _____________________________________________________________________________________________________________
> + More precision should be given about the way some noise is added to the data. A reference is given but one may wonder the impact of the way the noise is added on the performance of the algorithm, especially when using variational inference. Can you add a discussion on the impact of the choice of the noise model?
> + Can you detail how noise impacts precision and recall for each model?
> + What about the choice of prior on $Z$ and $Y$?
> + The claim is that the framework is general and that it can be combined with any method dealing with label noise. Can you explain the choice of co-teaching? What about the combination of the framework with other label-noise learning methods?
> _____________________________________________________________________________________________________________
> \
> \
> \
> **Q1.** More precision should be given about the way some noise is added to the data. A reference is given but one may wonder the impact of the way the noise is added on the performance of the algorithm, especially when using variational inference. Can you add a discussion on the impact of the choice of the noise model?
>
> **A1.** Thank you very much for raising this concern and the suggestion. The noise generation process is added exactly according to the causal graph in Fig. 1 of the main paper, i.e., the noisy label $\tilde{Y}$ is generated based on both $X$ and $Y$. Specifically, they fix the average flip rate to be $\rho$. Then, for each instance, they generate a transition matrix which summaries the flip rates of this instance. The instance with clean label y will be flipped according to the y-th row of the transition matrix. The implementation details can be found in Appendix B of [7].
> It is worth mentioning that the noise model in Fig.1 follows a very general setting, where the noise can be dependent on both instance $X$ and label $Y$. Therefore, our method can also be applied to instance-independent label noise (see Table 1 below for its performance in this setting), which is a special case of the considered setting (the causal from $X$ to $\tilde{Y}$ in the causal graph in Fig. 1 will disappear).
>
>
> |   | CausalNL| Co-teaching [1]| Decoupling [6]| MentorNet [3]| T-Revision [8]|
> |  -----------   |  ----------- |  -----------  |   -----------  |   -----------  |  -----------  |
> | pairflip-$45\\%$ | **82.39** $\pm$ 0.93 | 74.32 $\pm$ 1.27 | 56.03 $\pm$ 0.76 |  67.62 $\pm$ 0.58 | 75.14 $\pm$ 1.43|
>
> Table 1:  Classification accuracies (percentage) on CIFAR10 with instance-independent label noise pairflip-$45\\%$.
> \
> \
> \
> **Q2.** Can you detail how noise impacts precision and recall for each model?
>
> **A2.** Generally speaking, with an increasing noise rate, the precision and recall for each model should both decrease. Specifically, with the increasing noise rates for both positive and negative classes, true positives decrease, false positives increase, and false negative increases. Then for both precision and recall, the numerator becomes small, and the denominator hardly changes, finally leading to the decrease in both precision and recall.
> \
> \
> \
> **Q3.** What about the choice of prior on $Z$ and $Y$?
>
> **A3.** Prior on Z is chosen to be a Gaussian distribution with learnable parameters $\mu$ and $\sigma$, which is used by a number of VAEs (e.g., [2], and [4]). The nice property is that this will encourage the posterior $q(z|x)$ also to be a Gaussian and make it easier to sample a good latent representation. Prior on $Y$ is chosen to be a uniform distribution, which has been widely employed (e.g., [5], and [9]). Another advantage of using the uniform distribution is that it can prevent overconfidence of the learned class posterior.
> \
> \
> \
> **Q4.** Can you explain the choice of co-teaching? What about a combination of the framework with other label-noise learning methods?
>
> **A4.** Theoretically, without any constraints, the latent clean label Y is undefinable from noisy labels. To encourage the identifiability of $Y$, we could impose constraints on $P(Y|X)$. In our paper, we impose constraints on $P(Y|X)$ by a popular label-noise method known as co-teaching. Specifically, co-teaching implicitly selects confidence examples and uses them to approximate $P (Y|X)$.
>
> Furthermore, in general, most label-noise methods try to approximate $P(Y|X)$, and therefore, can be integrated into our framework. Two simple examples are that Decoupling [6] and MentorNet [3] can be directly integrated into our framework by simply replacing the co-teaching loss in Fig. 3 with their loss functions.
> \
> \
> \
> **References**
> _____________________________________________________________________________________________________________
> [1]. Han, Bo, et al. "Co-teaching: Robust training of deep neural networks with extremely noisy labels." arXiv preprint arXiv:1804.06872 (2018).
>
> [2]. Higgins, Irina, et al. "beta-vae: Learning basic visual concepts with a constrained variational framework." (2016).
>
> [3]. Jiang, Lu, et al. "Mentornet: Learning data-driven curriculum for very deep neural networks on corrupted labels." International Conference on Machine Learning. PMLR, 2018.
>
> [4]. Kingma, Diederik P., and Max Welling. "Auto-encoding variational bayes." arXiv preprint arXiv:1312.6114 (2013).
>
> [5]. Kingma, Diederik P., et al. "Semi-supervised learning with deep generative models." Advances in neural information processing systems. 2014.
>
> [6]. Malach, Eran, and Shai Shalev-Shwartz. "Decoupling" when to update" from" how to update"." arXiv preprint arXiv:1706.02613 (2017).
>
> [7]. Xia, Xiaobo, et al. "Part-dependent label noise: Towards instance-dependent label noise." Advances in Neural Information Processing Systems 33 (2020).
>
> [8]. Xia, Xiaobo, et al. "Are anchor points really indispensable in label-noise learning?." Advances in Neural Information Processing Systems 32 (2019): 6838-6849.
>
> [9]. Wu, Songhua, et al. "Class2simi: A noise reduction perspective on learning with noisy labels." International Conference on Machine Learning. PMLR, 2021.

---

> ### Author Response · Authors · 2021-08-28
> **Discussion**
>
> Dear Reviewer 38W1,
>
> We have tried our best to address all the concerns and provided explanations to all questions. If there are still unclear parts to you, please kindly let us know. We are very glad to further discuss them.
>
> Best,
>
> Paper5377 Authors

---

### Author Response · Authors · 2021-09-06
**Discussion**

Dear All,

Thanks for your constructive comments that help us improve this paper. We hope our rebuttal has cleared up the concerns you might have had. Please do not hesitate to let us know if there are any other concerns.

Many Thanks,

Paper5377 Authors

---

### Decision · Program_Chairs · 2021-09-27

**Decision:**

Accept (Poster)

**Comment:**

The paper considers the problem of label noise, but with the additional assumption that there is a causal relationship from the label to the instance (for example in image classification). By using a structural causal model, the authors propose a instance dependent noise model that shows good performance on benchmarks.

Three reviewers considered the paper, and all agree that the paper considers an important unsolved problem (instance dependent label noise), and provides a novel solution (via a structural causal model). The reviewers had several concerns which the authors satisfactorily addressed in their rebuttal. While one score was lower, the reviewer indicated in comments that they would support acceptance after the author rebuttal. A brief discussion post rebuttal among the reviewers resulted in all agreeing that the paper should be accepted. Therefore it is with great pleasure that I recommend the paper for NeurIPS.